# Wound Fluid from Breast Cancer Patients Undergoing Intraoperative Radiotherapy Exhibits an Altered Cytokine Profile and Impairs Mesenchymal Stromal Cell Function

**DOI:** 10.3390/cancers13092140

**Published:** 2021-04-29

**Authors:** Anne Wuhrer, Stefanie Uhlig, Benjamin Tuschy, Sebastian Berlit, Elena Sperk, Karen Bieback, Marc Sütterlin

**Affiliations:** 1Department of Obstetrics and Gynecology, University Medical Center Mannheim, Medical Faculty Mannheim, Heidelberg University, 68167 Mannheim, Germany; benjamin.tuschy@umm.de (B.T.); sebastian.berlit@umm.de (S.B.); marc.suetterlin@umm.de (M.S.); 2FlowCore Mannheim, Medical Faculty Mannheim, Heidelberg University, 68167 Mannheim, Germany; stefanie.uhlig@medma.uni-heidelberg.de (S.U.); karen.bieback@medma.uni-heidelberg.de (K.B.); 3Department of Radiation Oncology, University Medical Center Mannheim, Medical Faculty Mannheim, Heidelberg University, 68167 Mannheim, Germany; elena.sperk@umm.de; 4Institute of Transfusion Medicine and Immunology, Medical Faculty Mannheim, German Red Cross Blood Donor Services, Heidelberg University, 68167 Mannheim, Germany; 5Mannheim Institute for Innate Immunoscience, Medical Faculty Mannheim, Heidelberg University, 68167 Mannheim, Germany

**Keywords:** breast cancer, intraoperative radiotherapy, tumor microenvironment, mesenchymal stromal cells, oncostatin-M, IL-1β, leptin, VEGF, GROα, RANTES

## Abstract

**Simple Summary:**

Intraoperative radiotherapy (IORT) is increasingly used in the therapy of early breast cancer. Besides the direct radiotoxic effects, IORT may add at inhibiting local recurrences by a modulation of the microenvironment. Our aim was to assess the impact of IORT in altering immunological responses and wound healing processes. Thus, we analyzed surgical wound fluid collected after breast conserving surgery with and without IORT concerning acute changes in immune cell populations and cytokine levels. Furthermore, their impact on functions of breast cancer cells and mammary mesenchymal stromal cells (MSC) was assessed. We found no changes in the immune cell composition, yet group-related differences in the expression levels of several cytokines. The application of the wound fluid in MSC cultures caused group-dependent differences in MSC proliferation, wound healing and migration with an alteration of the MSC secretome. Our findings help to elucidate the biological effects of IORT and to clarify the concomitant role of MSC.

**Abstract:**

Intraoperative radiotherapy (IORT) displays an increasingly used treatment option for early breast cancer. It exhibits non-inferiority concerning the risk of recurrence compared to conventional external irradiation (EBRT) in suitable patients with early breast cancer. Since most relapses occur in direct proximity of the former tumor site, the reduction of the risk of local recurrence effected by radiotherapy might partially be due to an alteration of the irradiated tumor bed’s micromilieu. Our aim was to investigate if IORT affects the local micromilieu, especially immune cells with concomitant cytokine profile, and if it has an impact on growth conditions for breast cancer cells as well as mammary mesenchymal stromal cells (MSC), the latter considered as a model of the tumor bed stroma.42 breast cancer patients with breast-conserving surgery were included, of whom 21 received IORT (IORT group) and 21 underwent surgery without IORT (control group). Drainage wound fluid (WF) was collected from both groups 24 h after surgery for flow cytometric analysis of immune cell subset counts and potential apoptosis and for multiplex cytokine analyses (cytokine array and ELISA). It served further as a supplement in cultures of MDA-MB 231 breast cancer cells and mammary MSC for functional analyses, including proliferation, wound healing and migration. Furthermore, the cytokine profile within conditioned media from WF-treated MSC cultures was assessed. Flow cytometric analysis showed no group-related changes of cell count, activation state and apoptosis rates of myeloid, lymphoid leucocytes and regulatory T cells in the WF. Multiplex cytokine analysis of the WF revealed group-related differences in the expression levels of several cytokines, e.g., oncostatin-M, leptin and IL-1β. The application of WF in MDA-MB 231 cultures did not show a group-related difference in proliferation, wound healing and chemotactic migration. However, WF from IORT-treated patients significantly inhibited mammary MSC proliferation, wound healing and migration compared to WF from the control group. The conditioned media collected from WF-treated MSC-cultures also exhibited altered concentrations of VEGF, RANTES and GROα. IORT causes significant changes in the cytokine profile and MSC growth behavior. These changes in the tumor bed could potentially contribute to the beneficial oncological outcome entailed by this technique. The consideration whether this alteration also affects MSC interaction with other stroma components presents a promising gateway for future investigations.

## 1. Introduction

Breast cancer is the most frequent and the most fatal cancerous disease in women. Besides surgery, radiotherapy is an essential hallmark in the curative approach of therapy. After breast conserving surgery (BCS), adjuvant radiotherapy is the most important and most effective intervention to decrease the risk of intramammary relapses. Meta-analyses of randomized studies described a reduction of local and distant recurrence rates as well as a decreased breast cancer specific mortality [1,2]. Yet, 90% of all local relapses occur in direct proximity of the former tumor site [3], developing from cancer cells remaining in peritumoral tissue including perilymphatic and perivascular invasion or positive resection margins [4]. In recent years, the role of the tumor bed’s microenvironment within the progress of development of local recurrences has gained significance. Surgical tumor extirpation, although exerted with curative intent, causes wound healing and thus creates a local milieu that is not only beneficial for tissue regeneration, but also for local relapse and metastasis [5,6,7]. According to recent research, the induction of epithelial-to-mesenchymal transition (EMT) within the wound healing phase can enable epithelial tumor cells to gain invasive properties and thereby promote metastatic spread of the tumor [8,9].

The tumor bed presents itself as a promising target for local treatment forms of breast cancer. The technique of intraoperative radiotherapy (IORT) was developed to precisely apply high radiation doses directly into the wound cavity resulting from tumor extirpation, to eliminate residual tumor cells. Under clinical aspects, by this, the volume being most at risk to develop relapse is properly irradiated with a high single dose while sparing the surrounding tissue and delivering satisfactory cosmetic and toxicity results [10,11,12,13,14,15]. However, wound healing disorders are a possible side effect [16,17]. This suggests that IORT not only triggers a direct radiotoxic effect, but also affects tissue regeneration by modifying the local micromilieu.

We hypothesized that IORT provides an improved therapeutic intervention, not only by eliminating residual tumor cells, but also by altering the milieu within the tumor bed and the induced wound healing processes, involving immune and inflammatory responses. Especially the local immune response could convey the virtue of IORT in the treatment of a disease with early metastatic spread by an early antagonization of the formation of distant metastases [18,19], as IORT does not impair white blood counts during long-term follow-up [19]. In a previous investigation, we already pointed out that IORT efficiently targets the tumor bed by showing that mammary mesenchymal stromal cell (MSC) outgrowth does not occur after IORT [20]. This supports the notion that IORT targets cells of the tumor bed by modifying the growth conditions after BCS. Supporting this notion, Segatto et al. showed that IORT induced miR-223 expression in cells of the tumor bed. miR-223 expression led to reduced epidermal growth factor (EGF) expression which ultimately inhibited breast cancer cell growth and tumor recurrence in vivo [21].

Further, recent studies describe the influence of IORT on the cytokine composition of surgical wound fluid (WF) [22,23,24]. Belletti et al. made an essential contribution by examining the proteomic and functional properties of WF gained from wound drainages 24 h after BCS with and without IORT. IORT changed the molecular composition and the biological activity of WF and thus the growth conditions in the tumor bed, potentially to the disadvantage of recidivation [22]. Compared to WF from untreated patients, WF from IORT patients impaired the stimulatory activity on growth, invasion and migration of breast cancer cells in line with significant changes in IL-10 and IL-13 concentrations [22]. Kulcenty et al. further contributed to this by showing that IORT-mediated a radiation-induced bystander effect, which let to reduction of EMT and altered breast cancer cell gene expression and function [25,26]. To extend on this, we aimed to study the cytokine profile and the immune cell composition within the draining WF comparing IORT-treated and non-treated patients. To address the changes within the tumor milieu, we further assessed the effects of the WF not only on breast cancer cells but also on mammary MSC. Since MSC are part of the stroma and can be recruited into the tumor bed, the interaction of naive MSC with the irradiated tumor bed is of potential interest.

We hypothesized that IORT alters the tumor bed milieu targeting cells within the tumor bed (tumor cells and MSC) and also the cytokine milieu, by this not only modulating immune cell infiltration, but also tumor cell and MSC functioning. Thus, we examined surgical WF harvested 24 h after BCS from patients after BCS with and without IORT, analyzing (a) the cytokine milieu and (b) the composition of the immune cells. Furthermore, (c) the effect of WF on proliferation, wound healing and chemotactic migration of both breast cancer cells (cell line MDA-MB 231) and mammary MSC, isolated from biopsies of unirradiated patients, was assessed.

## 2. Materials and Methods

### 2.1. Patients

After having obtained informed consent (Ethics Committee II approval 2013-589N-MA), samples were collected from a total of 42 patients that were treated for breast cancer in the Department of Obstetrics and Gynecology of the University Medical Center Mannheim, Heidelberg University. Patients who received IORT during BCS formed the study group (= IORT group, 21 patients) and 21 women who underwent BCS without IORT formed the control group. In patients treated with IORT a single dose of 20 Gy (prescribed to the applicator surface) was delivered to the tumor bed during BCS in a standardized protocol (INTRABEAM^®^ System, Carl Zeiss Meditec, Oberkochen, Germany) immediately after the wide excision of the tumor [27]. Patients of the control group underwent BCS only without IORT. Both groups exhibited comparable compositions concerning molecular cancer phenotype and histological subtype (Table 1) without the presence of distant metastases.

### 2.2. Sample Collection

WF and peripheral blood was collected 24 h after surgery. WF was harvested from redon drainages inserted in the wound cavity. The WF samples were centrifuged, and the supernatants were cryopreserved at −80 °C. The cellular fractions of both sample types were processed for flow cytometric analysis as a standardized procedure.

### 2.3. Flow Cytometric Analysis

Isolated leukocytes of WF and peripheral EDTA blood samples were analyzed at the day of harvest. Absolute cell counts and counts of myeloid and lymphoid cells were determined using TruCOUNT^TM^ tubes (BD Biosciences, San Jose, CA, USA) and a simultaneous staining with the following mouse anti-human antibodies: myeloid panel: CD45, CD14, CD15, CD16, CD56, CD64; lymphoid panel: CD45, CD3, CD4, CD8, CD69, CD154 (fluorescence minus one (FMO) controls for CD69 and CD154); regulatory T cell (Treg) panel: CD4, CD25, CD127, CD196 and FoxP3 (all clones, fluorochromes and manufacturers, Appendix A). Gating was performed as shown in Appendix A. In addition, early and late apoptosis of white blood cells was measured using Annexin V/Propidium iodide (PI) dual staining [28]. All antibodies were properly titrated. Cells were acquired and analyzed using FACSCanto and FACSDiva software (Becton Dickinson, Franklin Lakes, NJ, USA).

### 2.4. Multiplex Cytokine Analysis

First a preliminary screening was performed. WF samples of the first 15 patients that were recruited (eight IORT vs. seven control) were individually analyzed using a semiquantitative human cytokine antibody array that detects 80 cytokines simultaneously (Human Cytokine Array C5, RayBiotech, Norcross, GA, USA). Evaluation was done with the open-source program Image J [29] and the corresponding Protein Array Analyzer tool [30]. Confirmatory ELISA tests were performed for HGF, OSM, GRO-α, IL-1β, Leptin, RANTES, uPA and VEGF (DuoSet, R&D Systems Inc., Minneapolis, MN, USA) on 19 IORT-vs. 20 control-WF samples. Due to limited sample volume, it was not possible to perform ELISAs on all 21 individuals per group (see specifications of associated figures).

### 2.5. Cell Culture

The breast cancer cell line MDA-MB 231 [22,31,32,33] stably expressing nuclear green fluorescent protein (GFP) was cultured using DMEM with 10% FBS and seeded at a standard density of 4000 cells/cm^2^.

MSC were isolated from unirradiated breast tissue biopsies as described [20]. MSC were cultured in DMEM (PAN Biotech, Aidenbach, Germany), 4 mM L-glutamine (Thermo Fisher, Waltham, MA, USA) and 10% human AB serum (German Blood Donor Service Baden-Württemberg-Hessen). The cells were cultured with a change of media twice weekly until 70–80% confluence. These cultures were passaged using trypsin-Ethylenediaminetetraacetic acid (EDTA, AppliChem, Darmstadt, Germany) 1× and cells were seeded at a standard density of 200 cells/cm^2^. A pool of five patients’ MSC exhibiting a representative marker profile was made and cryopreserved in fetal bovine serum (FBS)/10% dimethyl sulfoxide (DMSO, WAK Chemie Medical, Steinbach, Germany) in passage (P) 1.

### 2.6. Effects of Wound Fluid on Cell Function

The impact of WF on proliferation, wound healing and chemotactic migration of MDA-MB 231 cells and MSC was assessed using live cell imaging (IncuCyte ZOOM^®^ system, Sartorius, Hertfordshire, United Kingdom) (Appendix A). Cells were cultivated with increasing concentrations of pooled WF (0.1/0.5/1/5/10% IORT-WF vs. control-WF, pooled from all donors of the according group) supplemented to the culture medium. DMEM with 10% HS or FBS, respectively, was used for positive and serum-free DMEM for negative controls. All assays were reproduced in three independent experiments including multiple technical replicates. For quantification, we used adapted analysis masks (IncuCyte Basic Software, Sartorius, Hertfordshire, United Kingdom) (Appendix A).

Proliferation assay. MDA-MB 231 and MSC were seeded at a density of 2 × 10^3^ cells per well and cultivated at 37 °C, 5% CO_2_ for 24 h. MDA-MB 231 cells were cultured for additional 24 h in serum-free medium. Subsequently, the culture medium was replaced by WF-containing media. Cell proliferation was monitored using the IncuCyte ZOOM^®^ system and quantified as percent confluence using either phase contrast (MSC) or nuclear GFP (MDA-MB 231) values.Wound healing/scratch assay. MDA-MB 231 and MSC were seeded at a density of 2 × 10^4^ cells per well and incubated at 37 °C, 5% CO_2_ for 18 h. MDA-MB 231 cells were starved then for 24 h in serum-free medium. Afterwards, a scratch wound was set to the confluent monolayer (IncuCyte WoundMaker^TM^, Sartorius, Hertfordshire, United Kingdom). The cells were washed twice and WF containing media was added. Wound healing was monitored for 48 h. For both cell types, wound healing related migration was determined using phase contrast.Migration/chemotaxis assay. The insert plate of a IncuCyte ClearView 96 well plate (Sartorius, Hertfordshire, United Kingdom) was coated with fibronectin. Subsequently, MDA-MB 231 and MSC were seeded at a density of 1 × 10^3^ cells per well in DMEM + 1% HS or FBS; respectively. After settling for 1 h at ambient temperature, the medium was replaced with serum-free DMEM. The insert plate was mated with the reservoir plate loaded with WF-containing media. Cell migration to the reservoir plate was monitored for 48 h). For both cell types, chemotactic migration was determined using phase contrast.

### 2.7. Collection of Conditioned Media

To examine the influence of IORT-WF on the secretome of MSC, MSC were cultured in 0.5% pooled WF. After 72 h, conditioned media (CM) were harvested and cryopreserved at −80 °C. HGF, PDFG-BB, OSM, GRO-α, IL-1β, leptin, RANTES, uPA, I-309 and VEGF levels were assessed (DuoSet, R&D Systems Inc., Minneapolis, MN, USA) according to the manufacturer’s instructions.

### 2.8. Statistical Methods

Statistical tests were performed using JMP 13 statistical software (SAS Institute Inc., Cary, NC, USA). Data were calculated as the arithmetic mean ± standard deviation (SD). Statistical differences were calculated using double-sided *t*-tests for the cytokine and flow-cytometric analyses. For the live cell imaging analyses, Mauchly tests were conducted before MANOVA analyses to determine data sphericity. If there was a lack of data sphericity, Greenhouse-Geisser correction was applied. Differences were considered significant at *p* < 0.05.

## 3. Results

### 3.1. Immune Cell Subpopulations, Their Activation and Apoptosis State Are Not Changed in Wound Fluid from IORT-Treated Patients

To evaluate if IORT has an impact on the absolute count of immune cell subpopulations, their state of activation or their vitality, the cellular fraction of the surgical WF was analyzed and compared to peripheral blood samples taken from the patients. None of the examined subpopulations (myeloid cells, T cells, Treg) showed different cell counts or activation states comparing IORT vs. control 24 h after treatment (Figure 1A–C). Also, no group-dependent change in the apoptosis rates of the leucocyte fraction was apparent (Figure 1D). Therefore, no impact of IORT on cellular immune processes in the local environment could be determined within the first 24 h after BCS. Further, except for monocyte counts reduced from around 10% in peripheral blood to around 1% in WF, no significant differences in the cellular composition of WF and peripheral blood were found.

### 3.2. Wound Fluid from IORT-Treated Patients Exhibits an Altered Cytokine Profile

To ascertain whether humoral factors of the microenvironment are affected by IORT, cytokine levels in WF were assessed. Using a semiquantitative human cytokine array, we identified 30 cytokines with group-related changes (Figure 2A,B). GRO-α, IL-1β, Oncostatin-M and Leptin, all linked to tumor growth and inflammation, were significantly altered in the IORT group: Leptin increased; GRO-α, IL-1β, oncostatin-M decreased. More sensitive ELISA quantified the change to be 1.7-fold and 1.8-fold for IL-1β and leptin, respectively, and 0.6-fold for oncostatin-M (*p* < 0.05, Figure 3). GROα showed a similar trend, yet not significant. VEGF, RANTES, uPA and HGF were tested in addition given their potential role in the modulation of the local micromilieu [22,34,35,36,37]. Yet, IORT appeared not to affect the levels these factors in the WF (Figure 3).

### 3.3. Wound Fluid from IORT-Treated Patients Affects MSC Behavior

As IORT affects the levels of certain cytokines associated with tumor growth, we asked whether these changes cause functional alterations in tumor cells or MSC. First, the proliferation of MDA-MB 231 cells and MSC in increasing concentrations of WF (0.1%/0.5%/1.0%/5.0%/10.0%) versus 10% HS or FBS and serum free was assessed. We previously verified that WF in serum-free conditions can support proliferation. 0.5 and 1% WF were further evaluated as significant differences became apparent there.

In our experimental setup, proliferation of the MDA-MB 231 cell line was not changed comparing IORT-and control-WF. Also, wound healing and chemotactic migration were not affected (Figure 4).

We next addressed the WF effect on mammary MSC function [20]. We observed that WF from IORT-treated patients inhibits MSC proliferation with similar kinetics after 34 h, corresponding to the cell doubling time of MSC (30–40 h) (Figure 5). Cells treated with 0.5% control-WF displayed an increased proliferation rate thereafter (mean confluence at 70 h: 49.52% vs. 28.21% in IORT-WF, *p* < 0.0001). Proliferation of MSC treated with 1% IORT-WF was about 6 h delayed (mean confluence at 70 h: IORT 61.67% vs. control 50.48%, *p* < 0.0001).

Contrary to the MDA MB231 cell line, IORT-WF reduced not only MSC proliferation, but also wound healing, comparing 1% IORT-WF to control-WF (*p* = 0.01, Figure 5). Chemotactic migration towards 0.5% IORT-WF was also significantly reduced compared to control-WF.

All together these data show that IORT changes the composition of factors within the WF, which reduces MSC proliferation, wound healing capacity and chemotactic migratory activity.

### 3.4. Wound Fluid from IORT-Treated Patients Modifies the Secretome of MSC

Wondering about the significantly differing proliferation curve profiles of MSC in 0.5% IORT-vs. control-WF, we argued that the IORT-WF does not affect adhesion and spreading but mainly cell division. We suspected that the control-WF induced rise in proliferation may relate to MSC autocrine/paracrine factors, and that the IORT-WF may induce an anti-proliferative factor. Therefore, we analyzed the 72 h conditioned media from MSC cultivated in 0.5% IORT- and control-WF for the cytokines HGF, OSM, GRO-α, IL-1β, leptin, RANTES, uPA, I-309 and VEGF. Within the IORT-WF group, significantly reduced levels of RANTES, GROα and VEGF were found (Figure 6).

## 4. Discussion

IORT is expected to exert not only a direct radiotoxic impact on residual cancer cells in the irradiated tissue, but also to modify the local micromilieu in the tumor bed to hamper EMT and thus relapse. To address this, we assessed (a) immune cell subsets, activation state and apoptosis rates in WF and peripheral blood of control and IORT patients, (b) the cytokine composition in WF and (c) effects of WF on proliferation, migration and wound healing of the MDA MB 231 breast cancer cell line and mammary MSC.

### 4.1. Immune Cell Subpopulations, Their Activation and Apoptosis State Are Not Changed in Wound Fluid from IORT-Treated Patients

The composition of main lymphocyte subsets, their activation state and apoptosis rates within the draining WF and peripheral blood samples appeared not to be affected by IORT, when assessed 24 h post-surgery. This observation extends previous findings that the single application of IORT, in contrast to external radiation, does not affect white blood cell counts in peripheral blood samples with a follow-up period up to 4 years [19]. A recent work also reports unchanged frequencies in Treg, granulocyte and monocyte subsets, yet significantly increased CD56+high CD16+ NK cell numbers 3 weeks after IORT in peripheral blood [38]. The lower number of monocytes in WF in comparison to the peripheral blood samples may suggest their extravasation into the tissue as part of the local wound healing process [39]. Various previous studies reported that EBRT affects the innate and adaptive immune responses, and could induce an increased infiltration and activation of subsets of inflammatory cells into the tissue [40,41,42]. These changes were reported on tissues assessed after 48 h or later, while we assessed the WF samples standardized after 24 h. Given that our chosen panels did not involve B cells, future investigations could be expanded to B and more detailed innate immune and NK cell panels. Also, the identification of NK cells might be optimized by the usage of anti-CD3 in the lymphoid panel, since our strategy was the gating for positivity for CD45, CD56 and CD16 in addition to a homogenous positivity for CD64.

### 4.2. Wound Fluid from IORT-Treated Patients Exhibits an Altered Cytokine Profile

Confirming previous observations that IORT has an impact on cytokine levels within the WF [22,24], we observed that especially cytokines associated with wound healing and inflammation were modified by IORT. Oncostatin-M (OSM) was reduced in IORT-WF, whereas the levels of leptin and IL-1β were increased. These findings are in line with our hypothesis that IORT exerts an indirect anti-tumorigenic effect by changing the local micromilieu within the tumor bed. OSM is secreted by macrophages and cancer-associated adipose tissue and has been identified as a relevant factor in tumor progression [43]. Thereby, a selective inhibition of OSM resulted in a decreased peritumoral angiogenesis and an inhibition of the STAT3 signaling pathway [43]. Further, OSM suppressed the expression of estrogen receptor-alpha (ERα) of MCF 7 and T47D cell lines highly effectively and dose dependently [44]. Consequently, the estrogen receptor is not available as a target for anti-hormonal therapy, leading to a poorer overall prognosis. Furthermore, OSM has been shown to enhance the loosening of cell-cell and cell-matrix contacts of breast cancer cells and by this increasing invasiveness [45]. Beyond that, it acts as an external stimulus for EMT and as an inductor for cancer stem cells [46,47]. Since the signaling molecule Leptin is mainly produced by adipocytes, its high level in WF after IORT could possibly be attributed to radiation-induced damages of the fat tissue. Beyond the general known effect of Leptin as anti-hunger hormone, it has been shown to affect hematopoiesis, thermogenesis, reproduction and angiogenesis [48]. It exerts a slight pro-inflammatory effect and modulates the innate and adaptive immune response, linking metabolism and the immune system [48,49]. Leptin could explain the positive correlation between obesity and an elevated risk for breast cancer, accelerated tumor progression and poorer overall prognosis [50]. Yet the literature does not allow us to link the IORT-driven leptin-levels to pro- or anti tumorigenic effects.

The role of IL-1β during the pathogenesis of breast cancer appears to be similarly context-specific. One study describes IL-1β to trigger the development of osseous metastases [51], whereas other studies suggest that IL-1β could rather impede the colonization of metastasis inducing tumor cells [52]. The induction of inflammation processes by radiotherapy is commonly known, so that our finding of an elevated level of pro-inflammatory cytokines like IL-1β is plausible. Thus, we interpret IL-1β as a stress signal indicating the activation of the local immune system [53], which may assist to control tumor progression. Belletti et al. also observed significantly altered levels of the immune response-related cytokines IL-10 and IL-13 [22]. Elevated IL-1β levels were also reported by Kulcenty et al. [24]. However, they observed that, only WF of Luminal A breast cancer patients, but not that of the Luminal B subtype [24] showed IORT-mediated increased IL-1β levels. This fits to our observations, with the majority of samples representing the Luminal A subtype (Table 1).

### 4.3. Wound Fluid from IORT-Treated Patients Affects MSC, but not MDA-MB 231, Behavior

Belletti et al. previously showed that the local wound healing process affects the composition of the drainage WF and that this promotes proliferation, migration and invasion of breast cancer cells [22]. IORT, in contrast, nearly completely abrogated these effects. Contrary to their findings, we observed no changes of proliferation, wound healing and migration of MDA-MB 231 cells when incubated with different concentrations of IORT-WF. Arguing that the cells might be less dependent on external factors due to their highly malignant phenotype, we added an additional serum-starvation period for 24 h, but this was not sufficient to document potential differential effects of IORT-WF. We can also not exclude that the nuclear expression of GFP may have changed features of the MDA MB 231 cell line. Possibly, a three-dimensional culture would be needed to observe these effects: Belletti et al. showed that the differences between IORT- and control-WF were most notably in three-dimensional cell culture [22,54,55].

In a previous study, we already demonstrated that IORT ultimately restricts MSC outgrowth [20]. However, because MSC can become recruited to sites of injury or wound healing, we speculated that IORT could affect the micromilieu to reduce migration, wound healing and proliferation of MSC. Indeed, MSC’s functionality was reduced in the presence of IORT-WF compared to control-WF. Especially striking were the largely differing proliferation kinetics, suggesting an effect on cell cycle rather than adhesion. Postulating that IORT-WF affects the secretion profile of MSC’s autocrine signals [56], we detected significantly reduced levels of RANTES, GROα and VEGF. VEGF overexpression has been shown not to alter proliferation, morphology and differentiation of MSC [57], but whether reduced expression limits proliferation has to the best of our knowledge not been shown. Yet, all three factors are known to be related to tumor growth/tumor aggressiveness [58,59,60]. VEGF is an important therapeutic target to inhibit specifically the signaling pathway of angiogenesis and thus tumor growth [61]. GROα increased TNBC cancer cell migration and invasiveness, whereas a GROα knockdown diminished these effects. Moreover, corresponding signs of EMT were observed [62]. RANTES, also known as CCL5, plays a central role in the interaction of MSC with the tumor stroma and thus supports the mechanisms of metastasis of breast cancer [35]. A subcutaneously injected mixture of MSC and breast cancer cells of low malignancy formed a tumor xenograft with a significantly higher metastatic potential. Supporting our findings of an IORT-WF changed secretion profile of MSC, Kulcenty et al. reported similar radiation-induced bystander effects on breast cancer cell lines [23,24]. Conditioned medium of irradiated breast cancer cells exerted similar effects as IORT-WF: it reduced WF-mediated EMT and induced genotoxic effects [23,24]. Cells cultivated in IORT-WF showed transcriptomic changes: pathways related to cell proliferation, division, DNA damage response, and metabolism were enriched while inflammatory responses reduced [26]. Furthermore, the group showed that WF affected miRNA (miR-21, miR155 and miR-221] expression, yet differently in different breast cancer [63]. In three of four cell lines, WF reduced miRNA levels compared to control. In the HER2-overexpressing cell line, however, miRNA expression was strongly induced by control-WF, but to a lesser extent by IORT-WF. The three analyzed miRNA are associated with tumor progression and severity and may predict radiation responses.

In summary, besides showing that IORT appears not to induce short-term changes in the composition of immune cell subpopulations in WF and peripheral blood, we document changes of mammary MSC function and secretome. We hypothesize that these processes altogether contribute to the therapeutic efficacy of IORT, affecting the local cellular growth but also the auto-/paracrine tumor bed micromilieu in a way that is unfavorable for the development of local relapse and metastasis. With this new aspect, we contribute to a better understanding of how IORT influences the tumor bed by inducing a radiation-induced bystander effect.

## 5. Conclusions

Our results support the hypothesis that IORT affects components of the tumor bed not only by its radiotoxic effect but also by a radiation-induced bystander effect. Although not inducing short-term changes in the composition of immune cells, IORT modifies the local microenvironment indicated by changes in the cytokine content within the WF. This conditioning of the local environment by cytokines plays a decisive role within this context and may explain the IORT’s proven effectiveness. A better understanding of the direct and especially the indirect bystander effects may help to identify and improve effective treatments for breast cancer.

## Figures and Tables

**Figure 1 cancers-13-02140-f001:**
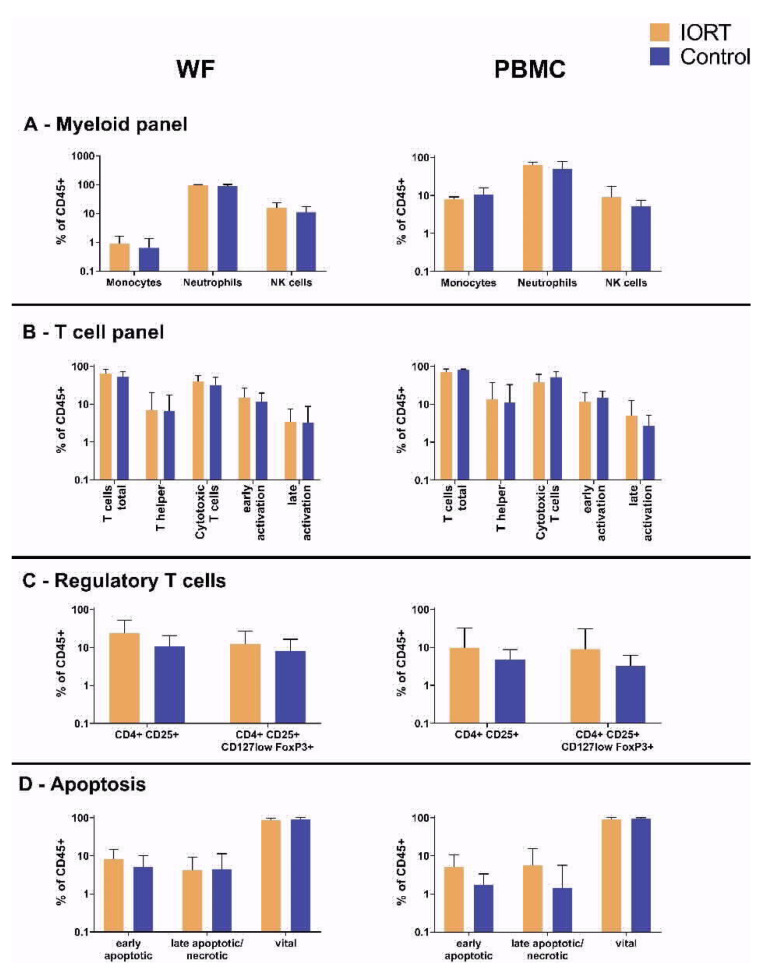
Absolute cell counts of myeloid (**A**), lymphoid (**B**), Treg (**C**) and apoptotic (**D**) cells. Indicated values: mean ± standard deviation, statistical analysis by double sided *t*-test. Myeloid analysis: *N*(WF) = 31 (IORT 19, control 12); *N*(PBMC) = 17 (IORT 11, control 6) Lymphoid analysis: *N*(WF) = 33 (IORT 21, control 12); *N*(PBMC) = 22 (IORT 15, control 7) Treg analysis: *N*(WF) = 24 (IORT 14, control 10); *N*(PBMC) = 16 (IORT 11, control 5) Annexin staining: *N*(WF) = 30 (IORT 20, control 10); *N*(PBMC) = 21 (IORT 14, control 7). (For PBMC, some patients did not consent to an additional blood sampling and for some patients, cell numbers were insufficient for analysis.).

**Figure 2 cancers-13-02140-f002:**
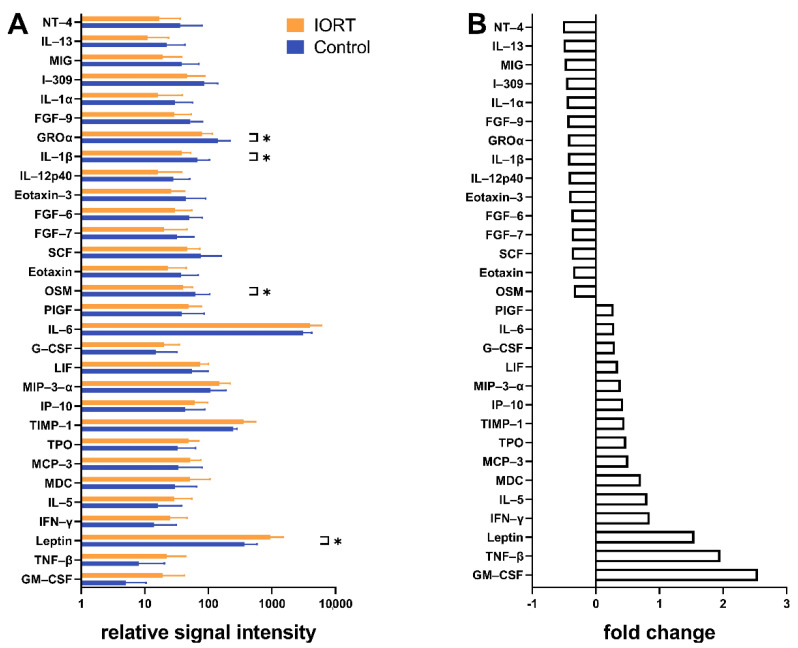
Signal intensities and calculated fold changes of the 30 cytokines with most differing levels between IORT and control group in the human cytokine array. (**A**): relative signal intensity, (**B**): fold change (>0: higher levels in IORT-WF; <0: higher levels in control-WF). *N* = 8 IORT vs. *N* = 7 control. Indicated values: mean ± standard deviation, statistical analysis by double sided *t*-test, * *p* < 0.05.

**Figure 3 cancers-13-02140-f003:**
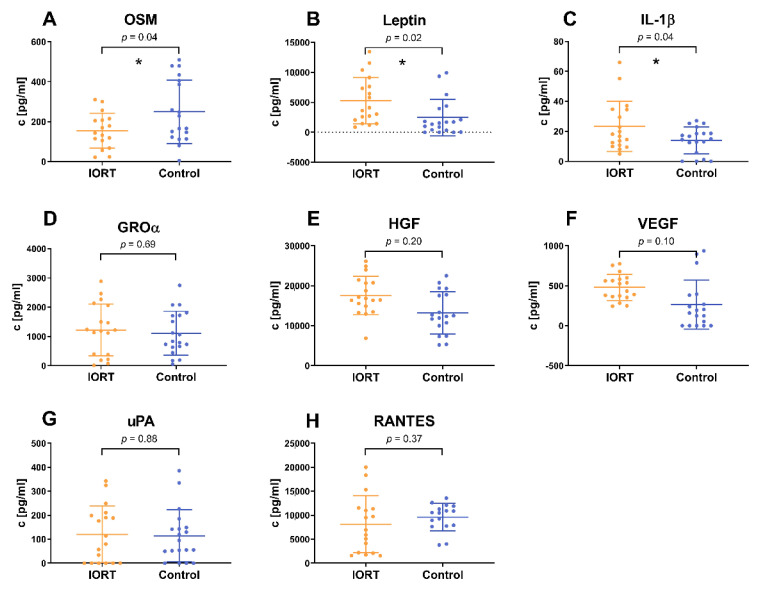
Cytokine concentrations tested in the individual WF samples by ELISA for OSM (**A**), leptin (**B**), IL-1β (**C**), GROα (**D**), HGF (**E**), VEGF (**F**), uPA (**G**) and RANTES (**H**). *N* = 18 IORT (yellow) and 18 control (blue), each *n* = 3. Indicated values: mean ± standard deviation, statistical analysis by double sided *t*-test, * *p* < 0.05.

**Figure 4 cancers-13-02140-f004:**
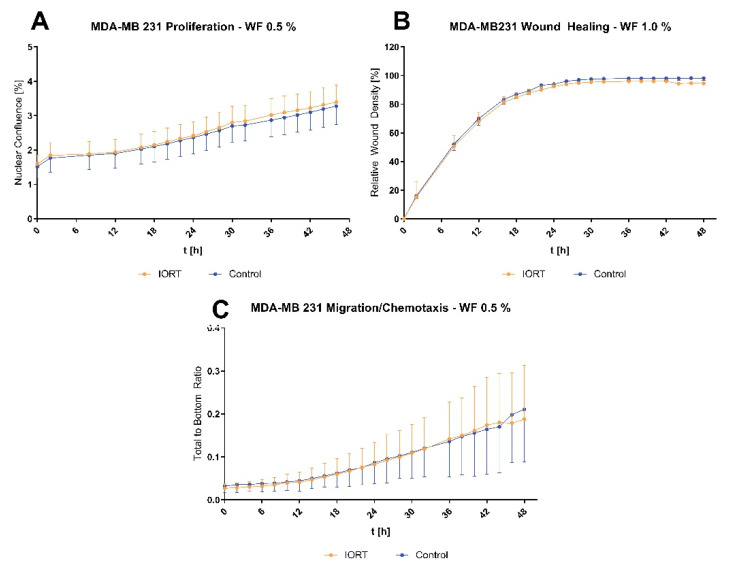
Functional analysis of MDA-MB 231 incubated with 0.5%/1% pooled WF concerning proliferation (**A**), wound healing (**B**) and chemotactic migration (**C**). All assays were reproduced in 3 independent experiments with each 8–12 technical replicates in each condition. Indicated values: mean ± standard deviation in percent over time. Statistical analysis by MANOVA.

**Figure 5 cancers-13-02140-f005:**
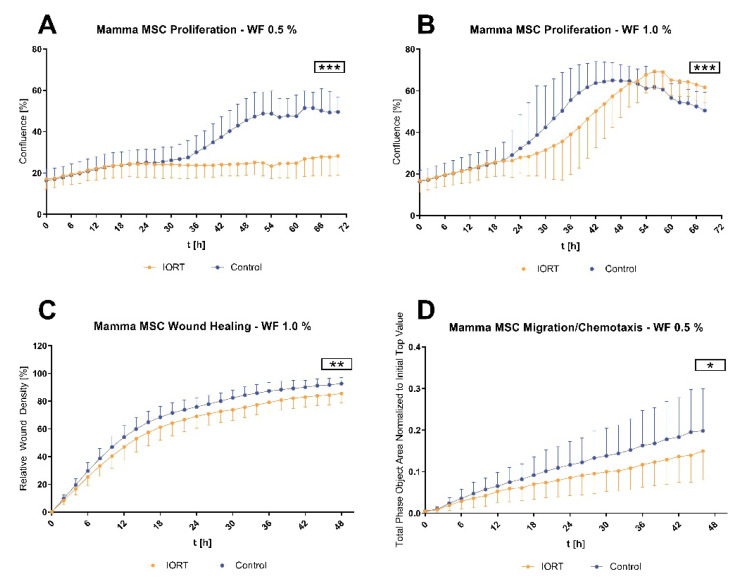
Functional analysis of MSC incubated with 0.5%/1% pooled WF concerning proliferation with 0.5 % WF (**A**), proliferation with 1.0 % WF (**B**), wound healing in 1.0 % WF (**C**) and chemotactic migration in 0.5 % WF (**D**). All assays were reproduced in 3 independent experiments with each 8–12 technical replicates for each condition. Indicated values: mean ± standard deviation in percent over time. Statistical analysis by MANOVA, * *p* < 0.05, ** *p* < 0.01, *** *p* < 0.0001.

**Figure 6 cancers-13-02140-f006:**
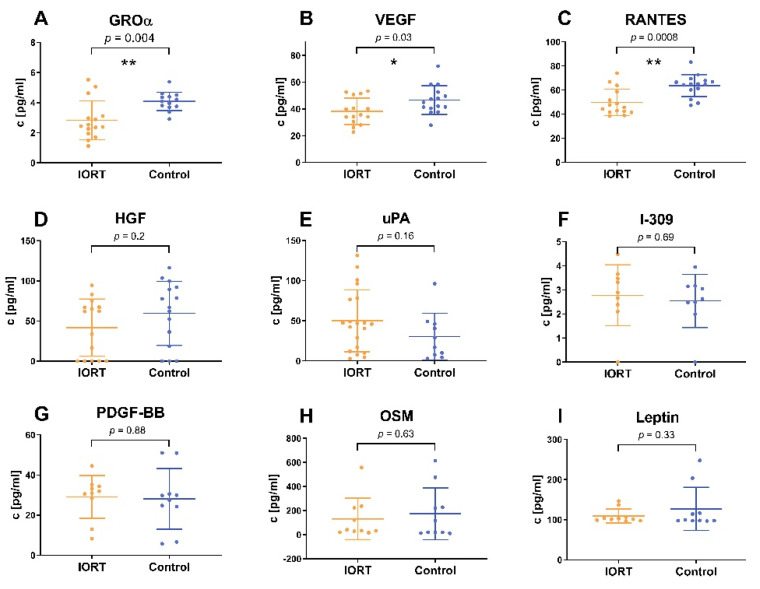
Cytokine concentrations in MSC conditioned media for GROα (**A**), VEGF (**B**), RANTES (**C**), HGF (**D**), uPA (**E**), I-309 (**F**), PDGF-BB (**G**), OSM (**H**) and leptin (**I**). All experiments were reproduced in three independent experiments, with each 8–12 technical replicates in each condition (IORT = yellow, control = blue). Indicated values: mean ± standard deviation, statistical analysis by double sided t test, * *p* < 0.05, ** *p* < 0.01).

**Table 1 cancers-13-02140-t001:** Distribution of histological subtypes and histological phenotypes in the tissue samples from patients of IORT and control group, *n* = 42 (IORT 21, control 21).

Specification	IORT	Control
- Histological Subtype		
No special type	81.0%	81.0%
Lobular histology	19.0%	14.3%
Tubular histology	0.0%	4.8%
- Molecular Phenotype		
Luminal A	57.1%	61.9%
Luminal B (HER2 negative)	33.3%	38.1%
HER2 positive	4.8%	0.0%
Triple negative	4.8%	0.0%

## Data Availability

The data presented in this study are available on request from the corresponding author. The data are not publicly available due to data protection and privacy.

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
