# Peer review of "Wound Fluid from Breast Cancer Patients Undergoing Intraoperative Radiotherapy Exhibits an Altered Cytokine Profile and Impairs Mesenchymal Stromal Cell Function"

_cancers, 2021, doi:10.3390/cancers13092140_

Round 1

Reviewer 1 Report

The manuscript falls under the scope of journal; It is an interesting topic.

The study presents the results of original research about IORT, local recurrence and microenvironment, local micromilieu, especially immune cells with concomitant cytokine profile.

The article is presented in an intelligible fashion; The work is described expansively.

The manuscript is written in standard English; however, there are some typos and the manuscript requires language editing prior to acceptance.

The abstract is appropriate with the study; The introduction is adequate however objectives and rationale of this study should be better presented.

Materials and Methods should be clearer; in the “Effects of wound fluid on cell function” the authors should describe better this section and add the correct references for it.

Statistical analysis are performed to a technical standard and are described in sufficient detail; Statistical tests were performed using JMP 13 statistical software. Data were calculated as the arithmetic mean ± standard deviation (SD). Statistical differences were calculated using double-sided t-tests for the cytokine and flow-cytometric analyses. Differences were considered significant at p < 0.05.

Results reported have not been published elsewhere.

The discussion should be improved: it is too long; it should be clearer and more linear. The Authors should describe better the benefits and issues due to the use of IORT.

Some bias and limits of this study (see number of cases: 42 breast cancer patients with breast-conserving surgery were included, of whom 21 received IORT (IORT group) and 21 underwent surgery without IORT (control group)) should be better underlined.

The conclusions are supported by the data but they should be better explained and presented in an appropriate fashion.

We also suggest to read and consider this reference:

Kulcenty K, Piotrowski I, Rucinski M, Wroblewska JP, Jopek K, Murawa D, Suchorska WM. Surgical Wound Fluids from Patients with Breast Cancer Reveal Similarities in the Biological Response Induced by Intraoperative Radiation Therapy and the Radiation-Induced Bystander Effect-Transcriptomic Approach. Int J Mol Sci. 2020 Feb 10;21(3):1159. doi: 10.3390/ijms21031159. 

The manuscript may be accepted with some revisions.

Author Response

Responses - Reviewer 1

The manuscript falls under the scope of journal; It is an interesting topic.

The study presents the results of original research about IORT, local recurrence and microenvironment, local micromilieu, especially immune cells with concomitant cytokine profile.

The article is presented in an intelligible fashion; The work is described expansively.

The manuscript is written in standard English; however, there are some typos and the manuscript requires language editing prior to acceptance.

Thank you very much for your comments. We thoroughly checked the manuscript and improved the language. These changes are not specifically marked.

The abstract is appropriate with the study; The introduction is adequate however objectives and rationale of this study should be better presented. Materials and Methods should be clearer; in the “Effects of wound fluid on cell function” the authors should describe better this section and add the correct references for it.

According to your proposal, we improved the presentation of objectives and rationale of the study and amended the referred sections of “Material and Methods”.

Statistical analysis are performed to a technical standard and are described in sufficient detail; Statistical tests were performed using JMP 13 statistical software. Data were calculated as the arithmetic mean ± standard deviation (SD). Statistical differences were calculated using double-sided t-tests for the cytokine and flow-cytometric analyses. Differences were considered significant at p < 0.05.

Results reported have not been published elsewhere.

The discussion should be improved: it is too long; it should be clearer and more linear. The Authors should describe better the benefits and issues due to the use of IORT. Some bias and limits of this study (see number of cases: 42 breast cancer patients with breast-conserving surgery were included, of whom 21 received IORT (IORT group) and 21 underwent surgery without IORT (control group)) should be better underlined. The conclusions are supported by the data but they should be better explained and presented in an appropriate fashion.

We have modified the text and highlighted the altered passages.

We apologize if we do not fully understand your comment regarding the number of cases. 21 patients per group is similar to the number of patients analysed in previous studies:

  • Belletti et al. DOI: 10.1158/1078-0432.CCR-07-4453: control – 25 patients vs. IORT (TARGIT) – 20 patients
  • Kulcenty et al. https://doi.org/10.3390/ijms21031159: control (BCS only) – 21 patients vs. IORT (BCS + IORT) - 22 patients),
  • Kulcenty et al. https://doi.org/10.3390/cancers12010011: control (BCS only) – 18 patients vs. IORT (BCS + IORT) luminal A and B subtypes – 20 patients
  • Kulcenty et al. DOI: 10.1038/s41598-019-44412-y: control = 21 patients IORT-boost 22 patients

We also suggest to read and consider this reference:

Kulcenty K, Piotrowski I, Rucinski M, Wroblewska JP, Jopek K, Murawa D, Suchorska WM. Surgical Wound Fluids from Patients with Breast Cancer Reveal Similarities in the Biological Response Induced by Intraoperative Radiation Therapy and the Radiation-Induced Bystander Effect-Transcriptomic Approach. Int J Mol Sci. 2020 Feb 10;21(3):1159. doi: 10.3390/ijms21031159.

We thank you for the suggestion, we integrated the reference.

The manuscript may be accepted with some revisions.

Reviewer 2 Report

Breast cancer is one of the most common cancer in women. Breast conserving surgery followed by radiotherapy is now a treatment of choice for a special types of breast cancer. However the recurrence as well as breast cancer specific mortality are decreasing each year, still around 90% of relapses occur in the same quadrant as the primary tumor. The surgical excision of the tumor causes wound healing, and thus creates a microenvironment that is not only beneficial for tissue regeneration, but also for local relapse and metastasis. The scientific data suggest, that modification of the tumor microenvironment may be beneficial in inhibiting local recurrences. Authors of this paper stated a question: can IORT have an impact on altering immunological response and wound healing process.

The topic is very interesting, and looking deeper in the changes of the tumor bed microenvironment after breast conserving surgery followed by IORT, may in future improve the treatment scheme.

In this paper authors performed cytokine profiling of wound fluids from patients after BCS and BCS followed by IORT, what is not new. Already others (Belletti 2008, Kulcenty 2019) have analyzed the WF from those groups finding different profile of cytokines between analyzed groups.  Moreover, the inflammation process as well as the impact of WF on BC behavior have already been published, and this papers should be mentioned in the text and cited (Belletti 2008, Segatto 2019, Zaleska 2016, Kulcenty 2019, 2020) The added value of this paper is the composition of immunological cells in WF and peripheral blood of BCS patients as well as changes of MSC secretom and proliferation after WF stimulation, which may have a high scientific soundness.

Minor revision:

line 170 – this reference that not correspond to MSC isolation

line 233 – better say: no significant apart from no apparent

Major revision

Fig. 1 – The analyzed PBMC control group is quite small (5-7 patients). There should be a proportion between analyzed and control group.

Fig. 4 – Analyzing the cell proliferation on BC cells, when the starting confluence is 1% and final 2% is without sense. How can results be interpreted after 48 hours with so little cells on plate? The huge SD in Fig. 4A may be dependent on that. The experiment should be performed again, with higher cell confluence. Based on this a question rises, at what confluence was the wound healing assay performed? Images should be added to see if the experiment was conducted correctly. On fig. 4A and 4C, the SD is very big, aa for the experiments based on established cell line, with pooled WF and such many technical replicates. Were the cells seeded correctly?

Concerning poor quality results, authors cannot take a conclusions of WF impact on BC cell line.

Concerning the BC cell line – why authors chosen this cell line? TNBC cancer, and especially this highly metastatic MDA-MB-231 cell lines does not correspond to BC patients treated with IORT. From authors collected WF, only 1 patient had TNBC phenotype. Some other cell lines corresponding to BC patients treated with IORT should be performed.

Author Response

Responses - Reviewer 2:

Breast cancer is one of the most common cancer in women. Breast conserving surgery followed by radiotherapy is now a treatment of choice for a special types of breast cancer. However the recurrence as well as breast cancer specific mortality are decreasing each year, still around 90% of relapses occur in the same quadrant as the primary tumor. The surgical excision of the tumor causes wound healing, and thus creates a microenvironment that is not only beneficial for tissue regeneration, but also for local relapse and metastasis. The scientific data suggest, that modification of the tumor microenvironment may be beneficial in inhibiting local recurrences. Authors of this paper stated a question: can IORT have an impact on altering immunological response and wound healing process.

The topic is very interesting, and looking deeper in the changes of the tumor bed microenvironment after breast conserving surgery followed by IORT, may in future improve the treatment scheme.

In this paper authors performed cytokine profiling of wound fluids from patients after BCS and BCS followed by IORT, what is not new. Already others (Belletti 2008, Kulcenty 2019) have analyzed the WF from those groups finding different profile of cytokines between analyzed groups.  Moreover, the inflammation process as well as the impact of WF on BC behavior have already been published, and this papers should be mentioned in the text and cited (Belletti 2008, Segatto 2019, Zaleska 2016, Kulcenty 2019, 2020) The added value of this paper is the composition of immunological cells in WF and peripheral blood of BCS patients as well as changes of MSC secretom and proliferation after WF stimulation, which may have a high scientific soundness.

Minor revision:

line 170 – this reference that not correspond to MSC isolation

We apologize, the reference number has been corrected.

line 233 – better say: no significant apart from no apparent

We changed the expression according to your suggestion.

Major revision

Fig. 1 – The analyzed PBMC control group is quite small (5-7 patients). There should be a proportion between analyzed and control group.

Unfortunately, patients of the control group gave less consent to an additional blood sampling the day after operation despite their initial approval. Yet, values of the control group PBMC were similar to values we obtained within other studies. Given the fact that we do not observe any differences between the groups 24h after BCS/BCS+IORT, we considered the differences of the group size annoying but not relevant for the data interpretation.

Fig. 4 – Analyzing the cell proliferation on BC cells, when the starting confluence is 1% and final 2% is without sense. How can results be interpreted after 48 hours with so little cells on plate? The huge SD in Fig. 4A may be dependent on that. The experiment should be performed again, with higher cell confluence. Based on this a question rises, at what confluence was the wound healing assay performed? Images should be added to see if the experiment was conducted correctly. On fig. 4A and 4C, the SD is very big, aa for the experiments based on established cell line, with pooled WF and such many technical replicates. Were the cells seeded correctly? Concerning poor quality results, authors cannot take a conclusions of WF impact on BC cell line.

We thank you for this comment. We have added a supplementary figure (S3) explaining the live cell imaging experiments: proliferation, scratch wound healing and chemotactic migration.

We used MDA-MB cells, which had a nuclear GFP expression. We have reported confluence based on this nuclear fluorescence. Compared to the overall cellular confluence, the nuclear confluence values are low, yet it allowed us to assess single cells, compared to the overall confluence which – at a certain density- could not discriminate single cells any more. The attached supplementary figure shows the comparison between the phase contrast-based confluence measurement and the green fluorescence nuclear confluence measurement.

As you may appreciate from the new supplementary figure, experiments have been performed using technical replicates (n= 4-12) and 3 independent experiments. Within each assay, paired analysis of control and IORT-WF was performed. Due to the highly sensitive assay, the apparently high standard deviation (mostly in the chemotaxis migration assay) results from the inter-assay variation from the three independent experiments. Thanks to your critical assessment, we re-checked the data and found a transmission error: In the preparation of our figures, we accidentally used the standard deviations of all experiments including pre-experiments with different seeding densities. This led to an apparently high standard deviation from the earliest time point on. We corrected the mistake, so that the current manuscript version displays the values of only the three independent experiments with comparable seeding densities. The statistical analysis was not affected by this transmission mistake. We apologize for the inconveniences. Furthermore, it is assay-related that the standard deviation in the chemotaxis migration assay gradually increases over time: both the non-migrated and migrated cells proliferate, this leads to an amplification of the standard deviation values with progressing time points.

Concerning the BC cell line – why authors chosen this cell line? TNBC cancer, and especially this highly metastatic MDA-MB-231 cell lines does not correspond to BC patients treated with IORT. From authors collected WF, only 1 patient had TNBC phenotype. Some other cell lines corresponding to BC patients treated with IORT should be performed.

We have specifically chosen this cell line to reproduce findings of Belletti et al. who showed that TARGIT treatment impairs the wound fluid-induced MDA-MB 231 migration (wound fluid pools at 2.5% concentration): https://clincancerres.aacrjournals.org/content/14/5/1325.

Likewise, Baldassare et al. reported a strong effect of IORT wound fluid on this cell line (proliferation, chemotactic migration and 3D motility): https://ascopubs.org/doi/abs/10.1200/jco.2007.25.18_suppl.21139

Martin et al. addressed our field stating that MSCs may promote breast cancer metastasis through facilitation of EMT, analyzing MDA-MB 231 in direct coculture with MSC:
https://link.springer.com/article/10.1007/s10549-010-0734-1

Furthermore, Mandel et al. observed the direct interaction of MSC with breast cancer cells using MDA-MB 231 for the in vitro experiments.
https://www.liebertpub.com/doi/full/10.1089/scd.2013.024
